# Quantum Cutting in KGd(CO_3_)_2_:Tb^3+^ Green Phosphor

**DOI:** 10.3390/nano13020351

**Published:** 2023-01-15

**Authors:** Dechuan Li, Jian Qian, Lei Huang, Yumeng Zhang, Guangping Zhu

**Affiliations:** 1School of Physics and Electronic Information, Huaibei Normal University, Huaibei 235000, China; 2Key Laboratory of Green and Precise Synthetic Chemistry and Applications, Ministry of Education, Huaibei 235000, China

**Keywords:** Gd^3+^, Tb^3+^, quantum cutting, phosphor

## Abstract

Phosphors with a longer excitation wavelength exhibit higher energy conversion efficiency. Herein, quantum cutting KGd(CO_3_)_2_:Tb^3+^ phosphors excited by middle-wave ultraviolet were synthesized via a hydrothermal method. All the KGd(CO_3_)_2_:*x*Tb^3+^ phosphors remain in monoclinic structures in a large Tb^3+^ doping range. In the KGd(CO_3_)_2_ host, ^6^D_3/2_ and ^6^I_17/2_ of Gd^3+^ were employed for quantum cutting in sensitizing levels. The excited state electrons could easily transfer from Gd^3+^ to Tb^3+^ with high efficiency. There are three efficient excited bands for quantum cutting. The excited wavelengths of 244, 273, and 283 nm correspond to the transition processes of ^8^S_7/2_→^6^D_3/2_ (Gd^3+^), ^8^S_7/2_→^6^I_17/2_ (Gd^3+^), and ^7^F_6_→^5^F_4_ (Tb^3+^), and the maximum quantum yields of KGd(CO_3_)_2_:Tb^3+^ can reach 163.5, 119, and 143%, respectively. The continuous and efficient excitation band of 273–283 nm can well match the commercial 275 nm LED chip to expand the usage of solid-state light sources. Meanwhile, the phosphor also shows good excitation efficiency at 365 nm in a high Tb^3+^ doping concentration. Therefore, KGd(CO_3_)_2_:Tb^3+^ is an efficient green-emitting phosphor for ultraviolet-excited solid-state light sources.

## 1. Introduction

Gd^3+^, one of the most popular ions, was used to sensitize Tb^3+^ [1,2,3,4], Eu^3+^ [5,6], Dy^3+^ [7,8], and Sm^3+^ [9,10] in luminescence emission. There are four excited energy bands, ^6^G*_J_*, ^6^D*_J_*, ^6^I*_J_*, and ^6^P*_J_*, below 5200 cm^−1^ in a Gd^3+^ ion [11]. Energy level matching enables excited electrons to transfer between different excited states by resonance, cross-relaxation, and phonon assistance [9,12]. The higher the energy of the exciting light, the greater the possibility of the energy level matching. In much of the literature, vacuum ultraviolet and short-wave ultraviolet are widely used for efficient quantum cutting [4,13]. In the energy range of 4500–5200 cm^−1^, the higher energy band is ascribed to Gd^3+^ absorption of the ^6^G*_J_* multiplet, which corresponds to the excitation light of vacuum or short-wave ultraviolet. Efficient emission of quantum cutting can be easily achieved with the assistance of ^6^G*_J_* [13,14,15,16,17,18]. When the excited state energy is lower than 3900 cm^−1^, energy level matching is difficult to achieve for quantum cutting. Some of the literature describes quantum cutting assisted with the ^6^D*_J_* and ^6^I*_J_* energy bands in the Tb^3+^ emission. However, excited state ions can easily transfer from Gd^3+^ to Eu^3+^ and Dy^3+^ by nonradiative relation and energy transfer in the ^6^D*_J_* and ^6^I*_J_* energy bands [19,20]. Additionally, the ^6^I*_J_* energy band consists of six energy levels: ^6^I_15/2_ (36,725 cm^−1^), ^6^I_13/2_ (36,711 cm^−1^), ^6^I_11/2_ (36,525 cm^−1^), ^6^I_17/2_ (36,461 cm^−1^), ^6^I_9/2_ (36,231 cm^−1^), and ^6^I_7/2_ (35,878 cm^−1^), which correspond to the theoretical excitation wavelengths of 272.4, 272.5, 273.9, 274.4, 276.1, and 278.8 nm in Gd^3+^ [11], respectively. The Gd^3+^ absorption of the ^6^I*_J_* level matches well with the 275 nm LED solid-state light source, which can be used to break through the limitation of the mercury lamp for miniaturization and portability of the product.

Rare earth carbonate is an excellent optical material with high photoluminescence [21,22] and birefringence [23,24]. In this work, we reported on the quantum cutting of Gd^3+^ sensitized KGd(CO_3_)_2_:Tb^3+^ phosphor in a relatively lower lever of ^6^D*_J_* and ^6^I*_J_*. In the phosphor, three important quantum cutting levels, ^6^I*_J_* (Gd^3+^), ^6^D*_J_* (Gd^3+^), and ^5^F*_J_* (Tb^3+^), were systematically studied by the luminesce spectra, quantum yields, and decay curves. The detailed transfer processes of excited electrons between different energy levels are discussed.

## 2. Materials and Methods

KGd_1-*x*_(CO_3_)_2_:*x*Tb^3+^ (*x* = 0, 0.005, 0.05, 0.1, 0.3, 0.5, 0.7, 0.9, note as KGdC:*x*Tb^3+^) phosphors were synthesized via a hydrothermal method with the raw materials Gd(NO_3_)_3_·6H_2_O (99.99%) and Tb(NO_3_)_3_·6H_2_O (99.99%). First, the nitrates of Gd(NO_3_)_3_ and Tb(NO_3_)_3_ were dissolved into deionized water. Second, the mixture was added to the K_2_CO_3_ solution (0.55 mol/L) under vigorous stirring. Third, diluted nitric acid was used to adjust the pH value (9.5) of the mixture solution. Finally, the reaction solution was transferred to an autoclave and heated at 200 °C for 8 h. Then, the precipitate was washed with deionized water and ethanol three times. The phosphor was dried at 60 °C (40 min) for the final product.

The crystal structures of KGd(CO_3_)_2_:*x*Tb^3+^ were analyzed by the X-ray diffractometer in the range of 10–80° (PANalytical, Almelo, the Netherlands). The morphologies and energy dispersive spectrum (EDS) were imaged via cold field emission scanning electron microscopy (Regulus 8220, Hitachi High-Tech Co., Tokyo, Japan). The luminescent properties were measured by an FLS920 fluorescence spectrophotometer equipped with 450 W Xe-lamp (Edinburgh Instruments, Livingston, UK). Using BaSO_4_ as a reference, powder samples were placed in the PTFE powder vessels and slotted into the sample holder, and then the absolute quantum yields were measured by the integrating sphere within the FLS920 sample chamber in an indirect method. The lifetimes of KGd(CO_3_)_2_:*x*Tb^3+^ were tested by the 60 W microsecond flashlamp (Edinburgh Instruments, Livingston, UK).

## 3. Results and Discussion

### 3.1. Crystal Structures

Figure 1 shows the diffraction patterns of Tb^3+^-doped KGdC at *x* = 0.1 and 0.9. In the variation in Tb^3+^ doping concentration, the diffraction peaks of samples are almost the same. The crystal structures remain stable in a larger Tb^3+^ doping range. The most intense diffraction peak is located at the (110) crystal plane for all samples. All the diffraction patterns are consistent with the standard card of monoclinic KGd(CO_3_)_2_ (JCPDS:1-88-1422) [25]. As the Tb^3+^ concentration increased, the diffraction peak of the (110) crystal plane shifted to a large angle. With a similar ion radius, Gd^3+^ (1.05 Å) could be easily substituted by Tb^3+^ (1.04 Å) at an arbitrary proportion [26]. Pure phase KGdC:*x*Tb^3+^ phosphors were successfully synthesized without any second phase.

### 3.2. Morphology

Figure 2 shows the morphologies of KGdC:*x*Tb^3+^ with different Tb^3+^ doping concentrations. As shown in Figure 2a–f, most of the particles exhibit well-developed monoclinic crystal grains with a size of about 80–350 μm. When the Tb^3+^ doping concentration is increased, the morphology of grains changes slightly. All grains show good monoclinic structure in samples with different Tb^3+^ doping ratios. The size of monoclinic grains is relatively large, especially at *x* = 0.3. Although large grains are easy to fracture, the type of grains could be clearly identified. A small area of monoclinic grain was selected to demonstrate the existence of elements in KGdC:0.5Tb^3+^. In the element distribution maps, K, Gd and Tb were well-dispersed (Figure 2g–i). As shown in Figure 2j, the element content was similar to the original stoichiometric ratio.

### 3.3. Luminescence Spectra

In the low Tb^3+^ doping concentration, the interactions between the emission ions are weak, and the transfer process of the excited state electrons is easily traced. Figure 3 shows the luminescence properties of KGdC:*x*Tb^3+^ at *x* = 0.005. The efficiently excited wavelengths are clearly shown in Figure 3a for the 542 nm emission. On the excitation spectrum, the intense excitation peaks are located at 244, 253, 273, 306, and 312 nm, corresponding to the Gd^3+^ transitions from ^8^S_7/2_ ground state to ^6^D_3/2_, ^6^D_9/2_, ^6^I_17/2_, ^6^P_5/2_, and ^6^P_7/2_ excited states [11], respectively. Additionally, the weak excitation peaks are located at 351, 368, and 376 nm, corresponding to the Tb^3+^ transitions from ^7^F_6_ to ^5^L_9_, ^5^L_10,_ and ^5^G_6_ [27,28], respectively. However, the intense peaks of the entire exciting spectrum mainly consist in the absorption of Gd^3+^, which indicates a high transfer efficiency from Gd^3+^ to Tb^3+^. The contribution of Gd^3+^ is indicated by the emission spectra in Figure 3b. Under the excitations of 244, 253, 273, and 306 nm, most of the Gd^3+^ excited state electrons could transfer to Tb^3+^ with the typical luminescence emission of 542 nm. Another typical emission of Gd^3+^ at 312 nm could be observed in all spectra, which indicates that a fraction of high excited state electrons could depopulate to ground state through the ^6^P_7/2_ intermediate excited state.

As shown in Figure 4a, the peak intensities at 273 nm decreased rapidly with the increasing Tb^3+^ concentration in KGdC:*x*Tb^3+^ phosphor monitored at 312 nm. The excitation intensities at 244, 253, and 273 nm are intense in low Tb^3+^ doping. When *x* is larger than 0.005, the excited intensities at 244 and 253 nm almost disappear. Furthermore, the exciting intensity at 273 nm also decreases rapidly. The excited intensity quenching of Gd^3+^ indicates that most excited state electrons cannot release energy through Gd^3+^ itself in a 312 nm emission at a high Tb^3+^ concentration.

Figure 4b shows the variations in exciting spectra under different Tb^3+^ concentrations in KGdC:*x*Tb^3+^ phosphor monitored at 542 nm. At the lowest Tb^3+^ doping, the contribution to the Tb^3+^ emission is effective, with the intense absorption peaks of Gd^3+^ at 244, 253, 273, 306, and 312 nm. When the Tb^3+^ doping concentration is increased, the exciting peaks of short wavelength are slightly shifted to the longer wavelength from 244 to 249 nm, corresponding to the Tb^3+^ spin-allowed transition [29,30]. The other excitation peaks remain in the same position except for the variation in excitation intensity. The most obvious change is that the excitation intensities at 283 nm increased with the Tb^3+^ concentration, which corresponded to the abolishment of the Tb^3+^ spin-forbidden transition [22]. In the range of 300–380 nm, the exciting peaks were related to the f–f transition of Tb^3+^ [30]. Two excitation bands show strong absorption at high Tb^3+^ concentrations, which is suitable for the excitation of commercial 275 and 365 nm UV LED.

The typical emission intensities of KGdC:0.3Tb^3+^ at different excitation wavelengths are shown in Figure 4c. In the emission spectra of KGdC:0.3Tb^3+^, the 312 nm emission peak almost disappears for the excitation at 245, 273, and 283 nm. Meanwhile, the efficient emission intensities at 542 nm are about four times higher than those excited at 312 and 351 nm at *x* = 0.3. The relative emission intensities of KGdC:*x*Tb^3+^ at 542 nm are shown in Figure 4d. As seen in the curves, the emission intensities at 245, 273, and 283 nm are relatively high. Especially for 245 nm excitation, the medium intensity of exciting light emitted the strongest emission light, which indicated that a higher efficient transfer existed between Gd^3+^ and Tb^3+^. For the excitation wavelengths of 273 and 283 nm, the phosphors exhibited a higher emission intensity in a wide range of Tb^3+^ doping. Moreover, the excitation spectrum of the phosphor is continuous in the range of 273 to 283 nm, which effectively matches the excitation of the 275 nm LED. For the excitation wavelength at 351 nm, the green emission intensity of phosphors slightly increased with increasing concentrations of Tb^3+^ ions without any concentration quenching. In addition, the decrease in emission intensity excited at 312 nm indicates that there is concentration quenching at this energy level in a high Tb^3+^ doping concentration.

### 3.4. Energy Level Diagram

Figure 5 shows the energy level diagram for the transitions of excited state electrons in the emission process. First, 244 nm photons are absorbed by Gd^3+^ in the ^8^S_7/2_→^6^D_3/2_ transition [31]. One of the excited state electrons in ^6^D_3/2_ depopulates to ^6^P_7/2_ in a non-radiation way; then, the electrons continually transfer to the ^8^S_7/2_ ground state, and emit 312 nm photons. Owing to the similar energy levels of 40,851 cm^−1^ (Gd^3+^, ^6^D_3/2_) [11] and 40,749 cm^−1^ (Tb^3+^, ^5^K_8_) [27], the other parts of the excited state electron transfer to the neighboring Tb^3+^ by resonance vibration. Then, the cross-relaxation occurs between two Tb^3+^: ^5^K_8_ + ^7^F_6_→^5^D_4_ + ^5^D_4_ (process 1) [21]. Finally, two excited state electrons in ^5^D_4_ are stimulated by one excited photon for the quantum cutting of ^5^D_4_→^7^F*_J_* (*J* = 6, 5, 4, 3) with typical Tb^3+^ emission at 542 nm.

In the excitation of 253, 273, and 306 nm, the ground state electrons of Gd^3+^ are excited to ^6^D_9/2_, ^6^I_17/2,_ and ^6^P_5/2_ levels, respectively. One of the excited electrons returns to the ^6^P_7/2_ in a non-radiation way for the 312 nm emission. Most excited state electrons are transferred to Tb^3+^ with high transfer efficiency [32]. The similar energy gaps of ^6^D_9/2_-^5^F_3_ (3200 cm^−1^) and ^7^F_6_-^7^F_3_ (3270 cm^−1^) are beneficial to the ^6^D_9/2_ electronic cross relation for the accumulation of ^5^F_3_ excited state electron populations in process 2 [27]. Meanwhile, ^6^I_17/2_ (Gd^3+^) excited state electrons also transfer to the adjacent ^5^F_3_ (Tb^3+^) level by resonance migration. Then, ^5^F_3_ excited state electrons relax to the ^5^F_4_ level. In process 3, the energy gaps between ^5^F_4_-^5^D_4_ (14,835 cm^−1^) and ^7^F_0_-^5^D_4_ (14,842 cm^−1^) are close enough for the cross-relation: ^5^F_4_ + ^7^F_6_→^5^D_4_ + ^5^D_4_ [27]. This is also another effective way to realize quantum cutting in KGdC:*x*Tb^3+^ phosphor. As for the Gd^3+^, the excited state electrons in ^6^P_7/2_ reach the ^5^D_4_ level of Tb^3+^ by cross-relaxation and non-radiation transition in processes 4 and 5. Another exciting peak at 283 nm is attributed to the Tb^3+^ absorption from ^7^F_6_ to ^5^F_4_. The excited state electrons are directly transferred to the ^5^D_4_ level by cross-relaxation for the two-photon emission.

### 3.5. Quantum Yield

Figure 6 shows the quantum yields (QY) of KGdC:*x*Tb^3+^ (*x* = 0.05, 0.1, 0.3, 0.5, 0.7, 0.9) under different excitation wavelengths. The results show that there are three efficient excitation bands in the quantum cutting process of KGdC:*x*Tb^3+^. On the top curve, the initial value of QY reaches 132% at *x* = 0.05 under the 245 nm excitation. Moreover, under the optimal excitation wavelength, all QY values of KGdC:*x*Tb^3+^ are greater than 100% and the maximum value is 163.5% at *x* = 0.5. This QY value is close to those of Tb^3+^-doped Ca_9_Y(PO_4_)_7_ (157%) [4], K_2_GdF_5_ (177%) [33], and KY(CO_3_)_2_ (177%) [22]. On the middle and lower curves, the exciting efficiency at 273 nm is greater than that at 283 nm only at *x* = 0.05. Quantum yields are greater than 100% in the Tb^3+^ doping range of 0.3–0.7. As for KGdC:0.5Tb^3+^, the maximum QYs are 119 and 143% under the excitations at 273 and 283 nm, respectively. In addition, the series of weak excitation peaks caused by the f–f transition of Tb^3+^ ions is also worthy of discussion. In the range of 320–390 nm, 351 nm is the typical excitation wavelength for Tb^3+^ emission. QY values increase with the increasing Tb^3+^ doping concentration. Compared with the other luminescent hosts, phosphate [34], fluoroborates [35], and silicate [36], concentration quenching does not occur in the KGC host with a high Tb^3+^ doping concentration. The unique structure of [REO_8_] polyhedral enables the sufficient distance between adjacent Tb^3+^ to avoid energy loss [21,23]. The maximum QY was 91.44% for KGdC:0.9Tb^3+^. It is interesting to confirm that the highly efficient excitation bands of quantum cutting are broad in KGdC host material, which is beneficial for choosing an appropriate exciting source for the application of solid-state lighting [37].

### 3.6. Decay Curve

Figure 7a shows the decay curves of KGdC:*x*Tb^3+^ under excitation at 273 nm. The electronic depopulation of the ^5^D_4_ level is characterized by the decay curve. As shown in the figure, all decay curves could be fitted by the single exponential. The fitted lifetimes mainly originate from the decay process of excited state electrons at the ^5^D_4_ level. The values of a lifetime were shown in Figure 7b. In the excitation at 273 nm, the lifetime values of the ^5^D_4_ level first increase and then decrease with an increasing Tb^3+^ concentration. The dramatic change originates from the competition between the ^6^I_17/2_ (Gd^3+^) and ^5^F_3_ (Tb^3+^) levels. The lifetime value is small in a small number of Tb^3+^. When the Tb^3+^ doping concentration is increased, the energy transfers between Gd^3+^ and Tb^3+^ are efficiently enhanced [38]. The lifetimes increase with the increasing number of excited state electrons. When *x* is larger than 0.3, the high concentration of Tb^3+^ improves the cross-relaxation efficiency between two Tb^3+^. The rapid accumulation of excited state electrons decreases the energy lifetime of the ^5^D_4_ level. As for the excitation at 245 and 283 nm, there are two lower excitation intensities for the 312 nm emissions than that of 273 nm in low Tb^3+^ doping concentration (as shown in Figure 4a). The competition of excited state electrons between Gd^3+^ and Tb^3+^ is weak. The Gd^3+^ emission consumes a small number of excited state electrons, which results in a slight reduction in lifetime. When the Tb^3+^ doping concentration is further increased, the lifetime decreases rapidly in a shortened distance between Tb^3+^ ions [39].

## 4. Conclusions

Tb^3+^-doped KGd(CO_3_)_2_ phosphors with high quantum yields were prepared via a hydrothermal method. Tb^3+^ could substitute for Gd^3+^ in the KGd(CO_3_)_2_ crystal lattice at any proportion. In the Tb^3+^-doped KGd(CO_3_)_2_ material, Gd^3+^ is both an activated and sensitized ion. The energy transfer between Gd^3+^ and Tb^3+^ is highly efficient. Quantum cutting can be effectively triggered at three excitation bands: 245, 273, and 283 nm. Under excitation oat 245 nm, the quantum yield of KGdC:*x*Tb^3+^ phosphor could reach 132% at *x* = 0.05, and the maximal value is 163.5% in the quantum cutting process of KGdC:0.5Tb^3+^. The excitation bands at 273 and 283 nm are continuous, which is consistent with the emission wavelength of commercial 275 nm LED. The highly efficient KGdC:Tb^3+^ phosphor could be potentially applied to the ultraviolet-excited solid-state light sources.

## Figures and Tables

**Figure 1 nanomaterials-13-00351-f001:**
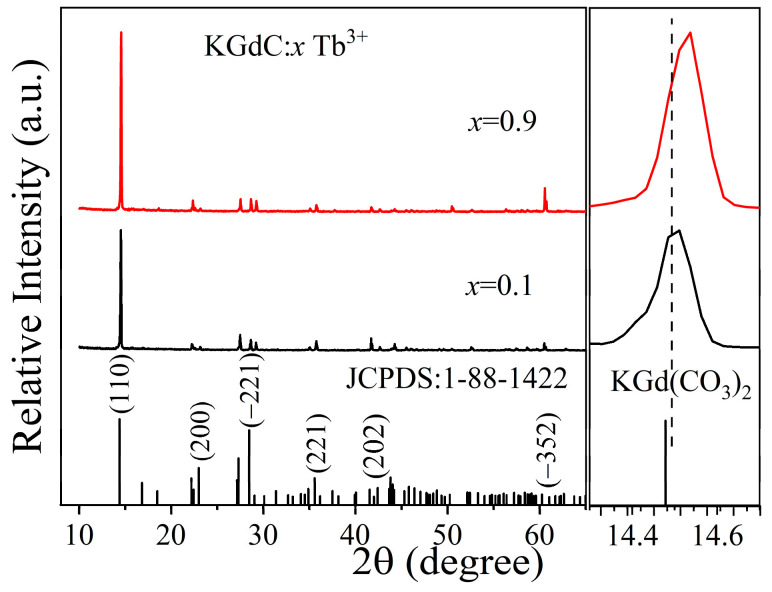
X-ray diffraction patterns of KGdC:*x*Tb^3+^ (*x* = 0.1, 0.9).

**Figure 2 nanomaterials-13-00351-f002:**
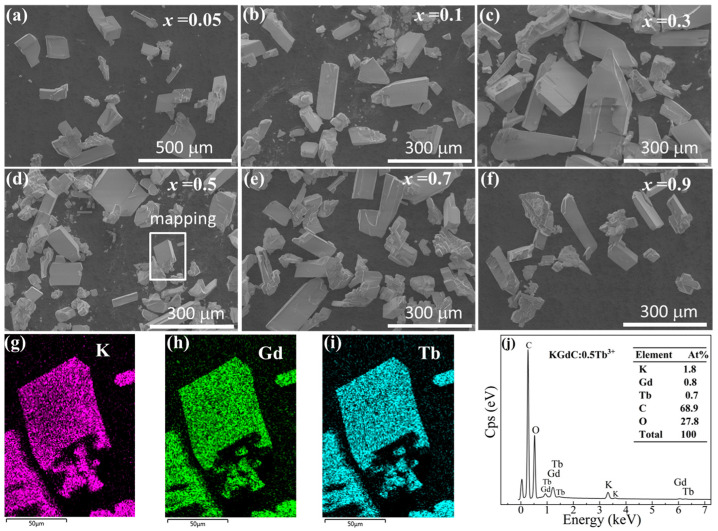
(**a**–**f**) Morphologies of KGdC:*x*Tb^3+^ (*x* = 0.05, 0.1, 0.3, 0.5, 0.7, 0.9); (**g**–**i**) Elemental mapping of K, Gd, and Tb; (**j**) Energy dispersive spectroscopy analysis.

**Figure 3 nanomaterials-13-00351-f003:**
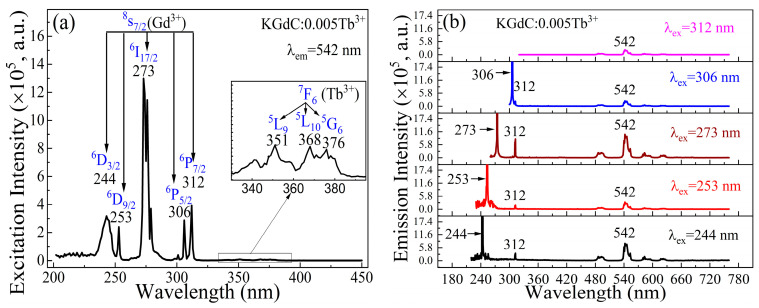
Luminescence properties of KGdC:0.005Tb^3+^: (**a**) Excitation spectrum; (**b**) Emission spectra excited at 244, 253, 273, 306, and 312 nm.

**Figure 4 nanomaterials-13-00351-f004:**
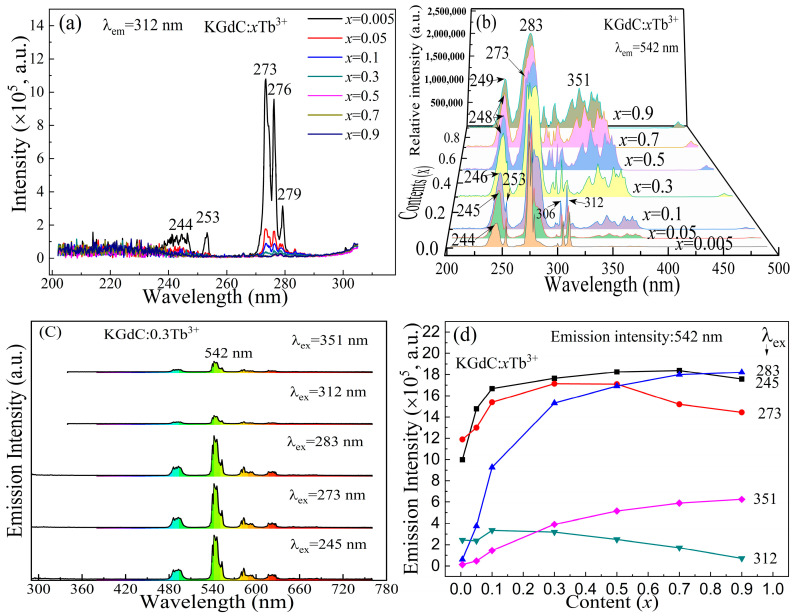
Luminescence properties of KGdC:*x*Tb^3+^ (**a**) Excited spectra monitored at 312 nm; (**b**) Excited spectra monitored at 542 nm; (**c**) Typical emission spectra of KGdC:0.3Tb^3+^ at different exciting wavelengths; (**d**) 542 nm emission intensities.

**Figure 5 nanomaterials-13-00351-f005:**
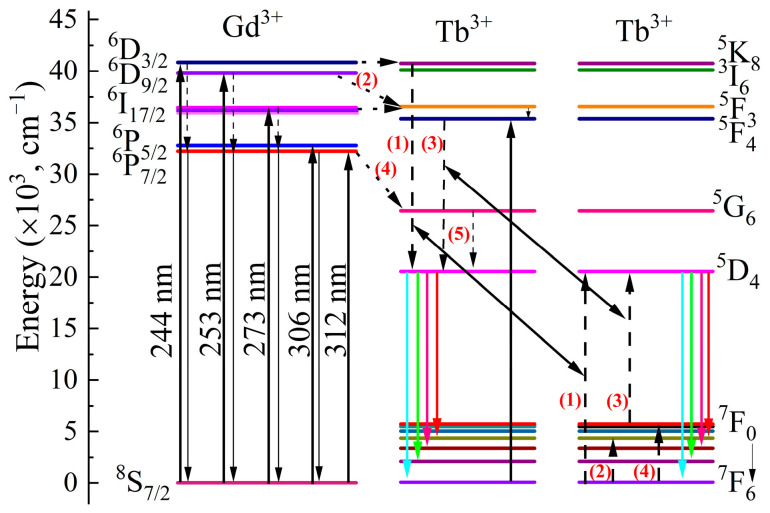
Energy level diagram of KGdC:Tb^3+^.

**Figure 6 nanomaterials-13-00351-f006:**
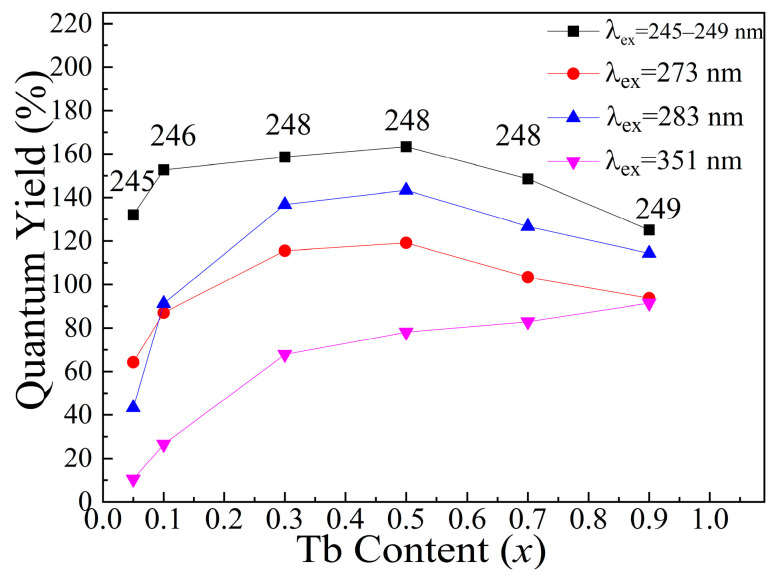
Quantum Yields of KGdC:*x*Tb^3+^ under different excitation wavelengths.

**Figure 7 nanomaterials-13-00351-f007:**
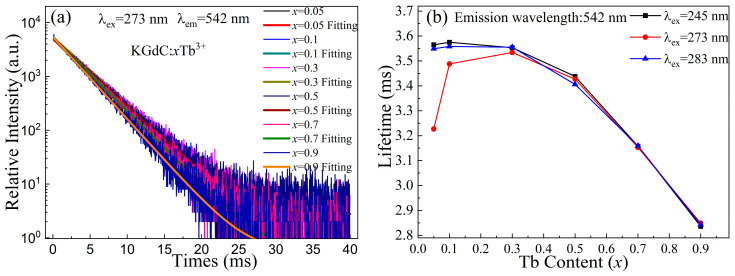
Decay properties of KGdC:*x*Tb^3+^ (**a**)decay curves; (**b**) Lifetimes of Tb^3+ 5^D_4_ level.

## Data Availability

Not applicable.

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
