# Peer review of "Quantum Cutting in KGd(CO_3_)_2_:Tb^3+^ Green Phosphor"

_nanomaterials, 2023, doi:10.3390/nano13020351_

Round 1
Reviewer 1 Report
The manuscript "Quantum cutting in KGd(CO3)2:Tb3+ green phosphor" by Dechuan Li has been presented nicely and interesting results were clearly discussed. For its publication there are certian changes to be implemented.
The standard XRD pattern and the data pattern seem to differ. The authors are hence suggested to have a Reitveld refinement not only for the presented 2 samples but also for all the samples in the study specifying the fit constant and the parameters necessary. The authors are thus encouraged to calculate the stress in the crystal system with the increase of dopant concentration.
What are the quatum yields, when the samples were excited directly to the Tb3+ ions i.e., at 351 nm excitation. The authors need to compare the QY of the samples when excited directly to Tb3+ ions for down shifting emission and to Gd3+ ions for down conversion (quantum cutting) emission.
The quantum yields of 150 % when excited to Gd3+ are quite surprising. In this sense do that authors ascertain to have QY of more than 75% when direct Tb3+ are excited? This result would enhance the quality of manuscript and explains the processes in deep.
The manuscript needs to be updated with the above suggestions and could be published.
Reviewer 2 Report
This is a well-organized paper. This paper provides an important contribution to the research on the development of KGd(CO3)2:xTb3+ phosphors and is potentially useful to readers in the preparation of an efficient green-emitting phosphor. After reading this paper from the beginning to the end, I cannot find serious weakness for this paper. I only have three comments to the authors, regarding the fundamental properties of this newly developed system.
First, throughout the paper, the authors claim to have synthesized new KGd(CO3)2:xTb3+ phosphors with high quantum yield. However, they do not present quantitative and qualitative studies to support the chemical structures of these phosphors. The XPS/ESCA, Raman spectroscopy and relative evidences of KGd(CO3)2:xTb3+ phosphors should be provided in the main text.
Second, based on current results, there are not clear the microstructural characterization of KGd(CO3)2:xTb3+ in the solid state. The TEM and SEM images of KGd(CO3)2:xTb3+ should be supplemented.
Third, there are many Tb-based phosphors reported in the literatures, and the authors have not proved the unique advantages of the present work when compared with those in the literatures.
Based on the suggestions above mentioned, this paper would be acceptable for Nanomaterials after revision.
Round 2
Reviewer 1 Report
The authors in their revision of the manuscript had implemented all the suggestions made in the first submission of the manuscript. Now, the manuscript can be accepted for its publication in the present form.
Reviewer 2 Report
After reading through the revised version of this manuscript and point-by-point response to reviewers' comments, the authors have addressed some of my common concerns and thus the paper can be suitable for publication in Nanomaterials.